# Resistance and Co-Resistance of Metallo-Beta-Lactamase Genes in Diarrheal and Urinary-Tract Pathogens in Bangladesh

**DOI:** 10.3390/microorganisms12081589

**Published:** 2024-08-05

**Authors:** Ayasha Siddique Shanta, Nahidul Islam, Mamun Al Asad, Kakoli Akter, Marnusa Binte Habib, Md. Jubayer Hossain, Shamsun Nahar, Brian Godman, Salequl Islam

**Affiliations:** 1Department of Microbiology, Jahangirnagar University, Savar, Dhaka 1342, Bangladesh; shantamicro44@gmail.com (A.S.S.); nahidulmicro44@gmail.com (N.I.); ronniebge22@gmail.com (M.A.A.); akterkakoli948@gmail.com (K.A.); marnusamomo@gmail.com (M.B.H.); nahar@juniv.edu (S.N.); 2Center for Health Innovation, Research, Action, and Learning—Bangladesh (CHIRAL Bangladesh), Dhaka 1205, Bangladesh; contact.jubayerhossain@gmail.com; 3Strathclyde Institute of Pharmacy and Biomedical Sciences, University of Strathclyde, Glasgow G4 0RE, UK; 4Division of Public Health Pharmacy and Management, School of Pharmacy, Sefako Makgatho Health Sciences University, Pretoria 0204, South Africa; 5Faculty of Medicine and Health, School of Optometry and Vision Science, UNSW Sydney, Sydney, NSW 2052, Australia

**Keywords:** Bangladesh, metallo-beta-lactamase, *bla*NDM-1, *bla*VIM, antimicrobial resistance, co-resistance, diarrhea, urinary tract infections, antimicrobial stewardship programs

## Abstract

Carbapenems are the antibiotics of choice for treating multidrug-resistant bacterial infections. Metallo-β-lactamases (MBLs) are carbapenemases capable of hydrolyzing nearly all therapeutically available beta-lactam antibiotics. Consequently, this research assessed the distribution of two MBL genes and three β-lactamases and their associated phenotypic resistance in diarrheal and urinary-tract infections (UTIs) to guide future policies. Samples were collected through a cross-sectional study, and β-lactamase genes were detected via PCR. A total of 228 diarrheal bacteria were isolated from 240 samples. The most predominant pathogens were *Escherichia coli* (32%) and *Klebsiella* spp. (7%). Phenotypic resistance to amoxicillin-clavulanic acid, aztreonam, cefuroxime, cefixime, cefepime, imipenem, meropenem, gentamicin, netilmicin, and amikacin was 50.4%, 65.6%, 66.8%, 80.5%, 54.4%, 41.6%, 25.7%, 41.2%, 37.2%, and 42.9%, respectively. A total of 142 UTI pathogens were identified from 150 urine samples. *Klebsiella* spp. (39%) and *Escherichia coli* (24%) were the major pathogens isolated. Phenotypic resistance to amoxicillin-clavulanic acid, aztreonam, cefuroxime, cefixime, cefepime, imipenem, meropenem, gentamicin, netilmicin, and amikacin was 93.7%, 75.0%, 91.5%, 93.7%, 88.0%, 72.5%, 13.6%, 44.4%, 71.1%, and 43%, respectively. Twenty-four diarrheal isolates carried *bla*NDM-1 or *bla*VIM genes. The overall MBL gene prevalence was 10.5%. Thirty-six UTI pathogens carried either *bla*NDM-1 or *bla*VIM genes (25.4%). Seven isolates carried both *bla*NDM-1 and *bla*VIM genes. MBL genes were strongly associated with phenotypic carbapenem and other β-lactam antibiotic resistance. *bla*OXA imparted significantly higher phenotypic resistance to β-lactam antibiotics. Active surveillance and stewardship programs are urgently needed to reduce carbapenem resistance in Bangladesh.

## 1. Introduction

Carbapenems are one of the last-line antibiotics prescribed for treating multidrug-resistant (MDR) Gram-negative bacterial infections [1]. As such, their prescribing should be carefully managed, assisted by designating them as ‘Watch’ antibiotics to be used with care and subject to antimicrobial stewardship programs (ASPs) to reduce antimicrobial resistance (AMR) [2,3]. However, there are concerns regarding the rising rates of AMR in Bangladesh, enhanced by sub-optimal healthcare standards and high levels of antibiotic misuse [4,5,6,7,8,9]. This includes appreciable purchasing of ‘Watch’ antibiotics without a prescription, adding to AMR despite current legislation, which is exacerbated by their ready availability in drug outlets throughout Bangladesh [10,11,12,13,14].

The emergence of carbapenemase-producing organisms (CPOs) with the capacity to hydrolyze carbapenem antibiotics has been reported worldwide [15,16,17], and is an increasing concern [18,19]. Initially *Klebsiella pneumoniae* was reported as the only carbapenemase-producing bacteria. Subsequently, other different bacterial species within the order of *Enterobacterales* were found with the competence of carbapenem hydrolysis and labeled as Carbapenem-Resistant Enterobacterales (CREs) [20]. However, notable heterogeneity exists in the mechanisms of CREs [18,21,22]. There are two main types of CREs, depending on the carbapenemase production capacity. These include carbapenemase-producing Enterobacterales (CP)-CRE and non-CP-CRE [23]. Earlier research identified that CP-CRE is more virulent than non-CP-CRE, and is associated with substantially higher mortality [18,23].

CPOs can be transmitted between patients more conveniently with the help of transferable integrons and mobile genetic elements (MGEs) than non-CPOs [21,24,25]. Biochemically, carbapenemases are categorized into two major groups. The first is serine-beta-lactamase, which includes serine at its active site and hydrolyzes the amide bond in the beta-lactam ring of antibiotics. The second category comprises metallo-β-lactamases (MBLs) enzymes containing the necessary zinc ions in their active sites that enable intended antibiotics to undergo hydrolysis [18]. Nine types of MBLs have been documented to date, of which three genes are globally prevalent, i.e., *bla*IMP, *bla*NDM, and *bla*VIM [26]. The MBL group antibiotic-resistance genes (ARGs) are increasingly being considered as emerging hazards and seen as significant threats to global public health [27].

In low- and middle-income countries (LMICs), diarrhea is a significant contributor to childhood mortality [28,29]. Pediatric diarrhea currently accounts for more than 1.7 billion cases annually and is the second leading cause of mortality among children aged five and below [30]. In 2017, it was estimated that pediatric diarrhea was responsible for over 533,000 deaths globally in children under five years of age, giving an estimated mortality rate of 78.4 (70.1–87.1) per 100,000 children [31]. Moreover, dehydration caused by diarrhea contributes significantly to approximately 1.5 to 2.5 million deaths annually among young children [30]. Consequently, the identification and appropriate management of pediatric diarrhea needs to be taken seriously.

The potential MDR in diarrheal bacteria arises from selective pressure from sustained antimicrobial exposure, especially antibiotics from the ‘Watch’ and ‘Reserve’ groups [14]. This complicates treatment, resulting in numerous antibiotics with limited potency. Consequently, treating the causative bacteria is becoming difficult and expensive [32]. Carbapenem-resistant hypervirulent diarrheal pathogens have now been reported in several countries as global emerging health threats [33,34], which needs to be reversed to reduce future morbidity, mortality, and costs associated with AMR [35,36,37,38]. The risks associated with AMR in LMICs are significant and need to be urgently addressed.

Urinary tract infections (UTIs) are among the most prevalent and endemic bacterial diseases globally [39,40]. Currently, approximately 40% of women experience a UTI episode at some point in their lives [41]. The commonly associated uropathogens are *Escherichia coli*, *Klebsiella pneumoniae*, *K. oxytoca*, *Citrobacter freundii*, *Pseudomonas aeruginosa* and *Serratia marcescens* [40]. However, increasing resistance to carbapenems among pathogens that cause common UTI infections has become an emerging problem [42,43,44].

All diarrheal pathogens and most UTI bacteria belong to the Enterobacterales order of Gram-negative bacteria that are increasingly resistant to carbapenem antibiotics in clinical and community settings worldwide [17,45]. The World Health Organization (WHO) has labeled CRE a global priority pathogen, considering its high transmission ability and hypervirulent emergence capacity [46]. Coordinated healthcare efforts and public health initiatives, including pertinent antimicrobial stewardship programs (ASPs) in ambulatory care, are imperative to prevent the transmission of CPO. Multiple studies have emphasized the need for continuous surveillance covering the epidemiology, phenotypic susceptibilities, and nucleic acid–based carbapenemase detection analyses. This information is crucial to instigate appropriate treatment strategies and control the spread of CPOs [47,48,49]. Such activities can be performed under the umbrella of ASPs [50,51,52]. However, concerns have been raised regarding the available resources and personnel to undertake ASP activities in LMICs [51,53]. This is now changing [54,55,56], although concerns remain regarding their implementation in Bangladesh [57].

The present study investigated the prevalence of two classical MBL genes, *bla*VIM and *bla*NDM-1, three extended-spectrum β-lactamase (ESBL) genes, *bla*TEM, *bla*OXA, and *bla*SHV and associated phenotypic resistance in pathogens isolated from diarrhea and UTI cases in Dhaka, Bangladesh. Other two ESBL genes, *bla*CTXM-9 and *bla*CTXM-15 were evaluated only in MBL-positive isolates. The findings can be used to guide future activities among all key stakeholder groups in Bangladesh, given the rising concerns in these priority groups. The study builds on the experiences in other countries [58].

## 2. Materials and Methods

### 2.1. Study Plan and Sample Collections

A longitudinal cross-sectional study was conducted from January 2020 to December 2020 to analyze carbapenem-resistance in diarrheal bacteria. Most study subjects were enrolled between April and August 2020, when diarrheal episodes appreciably increased in Bangladesh. Stool samples were collected from inpatients with acute diarrhea at the Uttara Adhunik Medical College Hospital in Uttara, Dhaka, Bangladesh. The samples were collected aseptically in sterile containers. The samples were subsequently transferred to the university laboratory that maintained cold storage facilities. The samples were then processed and cultured on the same day, to obtain the best results.

Urine samples were collected between April 2018 and March from inpatients admitted to Gonoshasthaya Samaj Vittik Medical College, located in Savar, and Gonoshasthaya Nagar Hospital in Dhaka. Patients were provided with sterile vials, and midstream urine samples were collected aseptically and stored at 4 °C before being transported to the laboratory. Urine samples were transported to the laboratory in an insulated ice box. The study patients also completed a standard questionnaire containing basic demographic data. The exclusion criteria included terminally ill and immunodeficient patients, particularly those with cancer, transplants, tuberculosis, kidney diseases, and HIV/AIDS.

### 2.2. Isolation and Identification of Bacteria

Diarrheal stools were diluted in phosphate-buffered saline and inoculated on Eosin-methylene blue–Levine (EMB-Levine) agar (Liofilchem Inc., Roseto degli Abruzzi, Italy), thiosulfate–citrate–bile salts–sucrose agar (TCBS) (Oxoid Ltd., Basingstoke, UK) and Salmonella–Shigella (SS) agar (Oxoid Ltd., Basingstoke, UK), simultaneously. Following inoculation, the agar plates were incubated for 18–24 h at 37 °C. The UTI samples were plated onto CLED medium (Liofilchem Inc., Roseto degli Abruzzi Italy) and MacConkey agar (Liofilchem Inc., Roseto degli Abruzzi, Italy) media. The plates were maintained under aerobic conditions at 37 °C for 24 h. For a specific type of pure culture, a single colony was taken from the culture plate and subsequently streaked onto Trypticase Soy agar (TSA) media and then incubated overnight at 37 °C. Bacterial isolates were first presumptively identified by various biochemical methods including Kligler Iron Agar (KIA), IMVIC (Indole, Methyl red, Voges–Proskauer and Citrate utilization), and catalase and oxidase tests. The API 20E kit (bioMerieux Inc., Craponne, France) kit was used to identify Enterobacterales and other Gram-negative rods. Bacterial identification was further confirmed by 16s rDNA sequencing (Macrogen Inc., Teheran-ro, South Korea).

### 2.3. Antimicrobial Susceptibility Test

The antimicrobial susceptibility patterns of the bacterial isolates were obtained using the Kirby–Bauer disk diffusion method. The classical β-lactam antibiotics, amoxicillin-clavulanic acid (30 μg), cefuroxime Sodium (30 μg), cefixime (30 μg), and cefepime (30 μg) were evaluated. For carbapenems, imipenem (10 μg) and meropenem (10 μg) were tested. Aztreonam (30 μg) was examined from the monobactam group. As aminoglycoside representatives, gentamicin (30 μg), netilmicin (30 μg), and amikacin (30 μg) were examined.

According to the standard guidelines provided by the Clinical and Laboratory Standards Institute (CLSI), a 0.5 McFarland turbidity standard was prepared. Subsequently, the suspensions were plated onto Mueller–Hinton agar (MHA) (Oxoid, Basingstoke, UK) media. Antibiotic discs were properly placed onto the media using sterile forceps, and the plates were subsequently placed in an incubator (37 °C) for overnight incubation. The sensitive bacteria were inhibited by the diffused antibiotics and developed a zone of clearance around the disc. The zone diameter was subsequently measured for evaluating antibiotic susceptibility patterns. *Escherichia coli* ATCC25922 was used as the susceptibility control reference strain for disc diffusion testing. The multiple antibiotic resistance (MAR) index was calculated and interpreted using the formula x/y, where ‘x’ represents the number of antibiotics an isolate was resistant to, and ‘y’ represents the total number of antibiotics tested [59].

### 2.4. Determining Minimum Inhibitory Concentrations (MICs)

After determining the antimicrobial susceptibility pattern, the agar dilution method was used to ascertain the minimal concentration of meropenem (MIC) needed to prevent bacterial growth. In this method, different concentrations of meropenem powder were used in MHA medium from 0.5 to 32 μg/mL. For the inoculum preparation, one pure culture colony was inoculated into TSB media and kept at 37 °C for two to three hours. To compare visually, a 0.5 McFarland standard was used (density 10^8^ CFU/mL). Plates were inoculated using a micropipette, and the inoculum was allowed to dry without moving the plates. The plates were inverted and placed in the incubator at 37 °C for 18 to 20 h. A control plate without meropenem was used to test the growth of the bacterial isolates. The *Etest* for meropenem was undertaken in parallel, using strips (Liofilchem Inc., Italy) carrying antibiotic concentration gradient (from 0.016 to 256 μg/mL) to validate the agar microdilution MIC results [60]. The results were interpreted for *Enterobacteriaceae* as per the CLSI guidelines, and the clinical breakpoints for meropenem resistance were considered when the MIC value was ≥4 μg/mL [61].

### 2.5. Molecular Detection of Metallo-Beta-Lactamase (blaNDM-1 and blaVIM) Genes

Polymerase chain reactions were used to detect two MBL genes, *bla*NDM-1 and *bl*VIM, in all the diarrheal and UTI bacterial isolates. The template bacterial DNA was extracted by the boiling method from bacteria cultured on nutrient agar media [62]. Specific primer sets for the respective MBL genes and ESBL genes were obtained from previous publications [63,64] and prepared by a commercial manufacturer (IDT Singapore Pte Ltd., Singapore). For a single PCR reaction, 12 μL 2X PCR pre-mixture (Thermo Fisher Scientific Inc., Carlsbad, CA, USA) was mixed with 2.0 μL of prepared bacterial DNA and five picomols of each primer (1 μL), and deionized water was added to make a final volume of 24 μL. Reactions went through 95 °C for 10 min for an initial denaturation, followed by 32 cycles of amplification by a thermal cycler (Applied Biosystems 2720, Singapore), following denaturation for 30 s at 94 °C, annealing for 30 s at 54–56 °C based on two primer sets, extension for 1 min at 72 °C, and a final extension of 7 min at 72 °C. The amplified PCR products were subjected to electrophoresis through 1.2% agarose gel at 100 volts for 30 min, subsequently stained with ethidium bromide, and visualized under UV light. A molecular weight standard was run alongside this, to measure the amplified PCR product sizes (GeneRuler, Thermo Fisher Scientific, Carlsbad, CA, USA).

### 2.6. Data Analysis

The SPSS statistical software package (version 25) and Graph Pad Prism (Version 9.5) were used to analyze the data. Descriptive and inferential statistical methods were utilized to describe diarrheal and UTI pathogens, the various metallo-beta-lactamase genes they carried, and their phenotypic characteristics. Pearson’s Chi-square test was used to test any associations between various categorical data. Statistical significance was determined by computing a two-tailed *p*-value (≤0.05).

### 2.7. Ethical Clearance

The Ethics and Research Review Committee of the Biological Sciences Faculty, Jahangirnagar University, approved the study of diarrheal pathogens [(BBEC/JU M2017 3(3), dated 15 March 2017] and UTI etiology [(BBEC/JU M2017 3(4), dated 15 March 2017]. Ethical permission from the JU was accepted among the three hospitals from which the samples were collected. All the research techniques conformed to the Declaration of Helsinki for medical research involving human subjects. For adult participants, urine and diarrheal samples were obtained with written consent. Written informed consent from legal guardians was obtained for those younger than 18 years old. Personal information was strictly protected.

## 3. Results

### 3.1. Study Population

A total of 240 diarrheal samples were obtained from patients with acute diarrhea attending the Uttara Adhunik Medical College Hospital. The clinical symptoms of the patients included 3–5 bowl movements with loose stools for the last 2–3 days, with or without fever, vomiting, and abdominal pain. Bacteria were detected in 228 samples. The overall detection rate of bacteria was 95.0% (228/240). The diarrheal episodes were well distributed by gender (134 males [56.0%] and 106 females [44.0%]). The age range was from three months to 80 years. The median age was one year. Further subclassification showed that children aged ten years or under accounted for over 80.0% of cases (Table 1).

A total of 150 urine specimens from patients with UTIs were collected from two hospitals in Dhaka and cultured on MacConkey agar and trypticase soy agar plates. Bacteria were detected in 142 samples, and the overall detection rate was 94.7% (142/150). Among these, 34 (24.0%) patients were male and 108 (76%) were female. The age of UTI-positive patients ranged from 4 to 76 years. The 10-year range of age classification identified the highest UTI infections in the 21–30-years age group (35.2%)**.** UTI cases were found to be higher among females (76.1%, 108/142) than among males (23.9%, 34/142). The detailed age distributions of the patients with diarrhea and UTI are shown in Table 1.

### 3.2. Bacterial Prevalence

Various bacterial species were isolated and identified from the collected diarrheal specimens. These included 102 *Escherichia coli* (44.7%), 20 *Klebsiella* spp. (8.8%), 19 *Escherichia fergusonii* (8.3%), 13 *Citrobacter freundii* (5.7%), and 15 *Enterobacter* spp. (6.6%), including *Enterobacter hormaechei*, *E. cloacae*, *E. mori*, and *E. tabaci*. Other low-frequency identified bacteria included *Shigella* spp., *Acinetobacter nosocomiali*, *A. radioresist*, *Aeromonas caviae*, *Bacillus cereus*, *Citrobacter europaeus*, *C. murliniae*, *C. pasteuri*, *Enterococcus faecium*, *E. gilvus*, *E. hirae*, *Klebsiella oxytoca*, *K. variicola*, *Proteus mirabilis*, *Pseudomonas parafulva*, *Serratia marcescens*, *Staphylococcus warneri*, *Staphylococcus epidermidis*, and *Vibrio neocaledonicus*.

Various bacteria were also isolated from urine samples dominated by 56 *Klebsiella* spp. (39.4%), and 34 *Escherichia coli* (23.9%), in addition to 17 *Enterobacter* spp. (12.0%), 14 *Proteus* spp. (9.9%), 14 *Pseudomonas* spp. (9.9%), and 5 *Staphylococcus* spp. (3.5%). Other identified UTI pathogens included *Acinetobacter pittii*, *Enterobacter cloacae*, *E. hormaechei*, *Proteus mirabilis*, and *Pseudomonas putida* (Appendix A).

### 3.3. Phenotypic Resistance and Minimum Inhibitory Concentration

Resistance to β-lactam and aminoglycoside antibiotics was observed in 228 isolates obtained from the diarrheal samples. Among the isolates, 50.4% showed resistance to amoxicillin-clavulanic acid, 65.5% to aztreonam, 66.8% to cefuroxime, 80.5% to cefixime, 54.4% to cefepime, 41.6% to imipenem, 25.7% to meropenem, 41.2% to gentamicin, 37.2% to netilmicin and 42.9% to amikacin. Meropenem showed maximum activity, whereas cefixime, a third-generation cephalosporin, was the least active.

Among the 142 UTI isolates, 93.7% showed resistance to amoxicillin + clavulanic acid, 75.0% to aztreonam, 91.5% to cefuroxime, 93.7% to cefixime, 88% to cefepime, 72.5% to imipenem, 13.6% to meropenem, 44.4% to gentamicin, 71.5% to netilmicin and 43% to amikacin. For the UTI isolates, meropenem also showed the highest potency, while amoxicillin plus clavulanic acid and cefixime were the least active antibiotics. Overall, UTI pathogens exhibited significantly higher resistance (*p* = 0.001) against the β-lactam antibiotics and aminoglycoside antibiotics tested, except for meropenem (Figure 1A). Significantly higher resistance to meropenem in the disk-diffusion test in diarrheal pathogens (*p* = 0.001) showed good agreement with the meropenem-MIC assay (Figure 1B). The isolates exhibited varying degrees of meropenem MIC, ranging from ≤0.5 to ≥32 µg/mL. Most of the isolates were sensitive (MIC, ≤1 µg/mL). Pathogens with MIC values of ≥4 µg/mL were considered resistant. MIC value ≥ 32 µg/mL of meropenem was found in over 21% of diarrheal pathogens compared to 5% of UTI pathogens (Figure 1B, *p* = 0.001). Overall, in 370 clinical isolates, resistance to amoxicillin-clavulanic acid, aztreonam, cefuroxime, cefixime, cefepime, imipenem, meropenem, gentamicin, netilmicin and amikacin was 66.8%, 68.4%, 75.9%, 85.1%, 67.1%, 53.2%, 20.8%, 42.2%, 50%, and 42.7%, respectively. A total of 13.2% of isolates exhibited a meropenem MIC value ≥ 32 µg/mL.

### 3.4. Prevalence of MBL and ESBL Genes

*bla*NDM-1 and *bla*VIM were identified in 6.8% (25/370) and 9.5% (35/370) of the total bacteria analyzed. Among the 228 diarrheal isolates, *bla*NDM-1 and *bla*VIM were detected in 12 (5.3%) isolates. One of the isolates carried the *bla*NDM-1 and *bla*VIM genes together. Among the 142 UTI isolates, *blaNDM*-1 was detected in 13 (9.2%) isolates, and *blaVIM* was detected in 23 (16.2%) isolates. Seven UTI isolates carried both the MBL genes together. Overall, a significantly higher prevalence of the two MBL genes was identified in UTI isolates compared to diarrheal pathogens (*p* = 0.001). The ESBL gene, *bla*TEM, was identified in 44.6% (165/370) of the isolates, 90 (39.5%, 90/228) from diarrheal isolates, and 75 (52.8%, 75/142) from UTI isolates (*p* = 0.013). Overall, *bla*OXA was identified in 26.2% (97/370) of the isolates. Likewise, UTI isolates carried a significantly higher prevalence of blaOXA than was seen in the diarrheal isolates (46.5% versus 13.6%, *p* = 0.000). The total frequency of *bla*SHV was comparatively lower (8.1%, 30/370), and no significant differences were observed between the isolates of diarrhea and UTI (8.3% versus 7.7%, *p* = 1.0).

The overall ESBL coexistence covering *bla*TEM, *bla*OXA, *bla*SHV, *bla*CTXM-9, and *bla*CTXM-15 genes was identified in 69.2% (36/52) of MBL-producing isolates. The existence of the *bla*TEM was the most prevalent in MBL-producing isolates, at 46%, followed by *bla*OXA (42.3%), *bla*CTXM-15 (39.1%), *bla*CTXM-9 (13%) and *bla*SHV (7.7%). The coexistence of the β-lactam genes was found in other different combinations in MBL-positive strains. The *bla*TEM was found to coexist with *bla*OXA-type variants in 9.6% (5/52) of the MBL producers. The same prevalence (9.6%) was found in the combination of *bla*TEM and *bla*CXTM-15 types. The coexistence of four genes, *bla*TEM, *bla*OXA, *bla*SHV, and *bla*CXTM-15 was observed in two isolates. The seven isolates (13.5%) that carried both *bla*VIM and *bla*NDM-1 were also found to coexist with *bla*TEM and *bla*OXA.

### 3.5. Association of Phenotypic and Genotypic (blaNDM-1 and blaVIM Gene) Resistance

A total of 25 *bla*NDM-1-positive (6.8%), and 35 *bla*VIM-positive (9.5%) clinical isolates were identified. Both the two MBL genes showed significantly higher resistance association against imipenem and meropenem. However, we found no statistical associations between the phenotypic resistance of the β-lactam antibiotics and the presence of the two MBL genes (Appendix A).

There were 228 diarrheal isolates, of which 12 isolates were *bla*NDM-1 positive and the other 12 were *bla*VIM positive. Of the 12 *bla*NDM-1 positive isolates, 67% were resistant to amoxicillin-clavulanic acid and imipenem, 58.2% to aztreonam, 50% to cefuroxime and cefepime, 83% to cefixime, 41.7% to meropenem, gentamicin, and amikacin, and 25% to netilmicin. However, of the 216 *bla*NDM-1 negative isolates, 49.5% were resistant to amoxicillin-clavulanic acid, 66% to aztreonam, 68% to cefuroxime, 80% to cefixime, 55% to cefepime, 40% to imipenem, 25% to meropenem, 41% to gentamicin, 38% to netilmicin and 43% to amikacin. We detected a very weak association between the phenotypic resistance of some beta-lactam and aminoglycoside antibiotics and the presence of *bla*NDM-1 (aztreonam, cefuroxime, cefixime, cefepime, gentamicin, and netilmicin). The resistance percentages between *bla*NDM-1 positive and *bla*NDM-1 negative pathogens appeared almost equal against these antibiotics. However, the association was quite strong in the case of amoxicillin-clavulanic acid, imipenem, meropenem, and amikacin (Appendix A).

Among the 12 *blaVIM-positive* diarrheal isolates, 50% were resistant to amoxicillin + clavulanic acid and cefepime, 58% to aztreonam, 67% to cefuroxime, cefixime, and gentamicin, 75% to imipenem, 41.7% to meropenem, 25% to netilmicin, and 33% to amikacin. Of the 216 *bla*VIM-negative isolates, 50.5% were resistant to amoxicillin-clavulanic acid, 66% to aztreonam and cefuroxime, 81.3% to cefixime, 54.7% to cefepime, 39.7% to imipenem and gentamicin, 24.8% to meropenem, 38% to netilmicin and 43.5% to amikacin. Consequently, the absence or presence of the *bla*VIM gene in diarrheal isolates had only a limited effect on the resistance to most of the antibiotics used in this study, except imipenem (*p* = 0.05), meropenem (*p* = 0.25) and gentamycin (*p* = 0.17) (Appendix A).

There were 142 UTI isolates, of which 13 isolates were *bla*NDM-1-positive, and the other 23 isolates were *bla*VIM-positive. Among the 13 blaNDM-1-positive isolates, all were resistant to amoxicillin-clavulanic acid, cefuroxime, cefixime, cefepime, gentamicin, netilmicin and amikacin, 84.6% to imipenem, 77% to aztreonam and 69% to meropenem. *bla*NDM-1 negative isolates also showed similar levels of resistance to amoxicillin-clavulanic, aztreonam, cefuroxime, cefixime, cefepime, and imipenem. Amoxicillin-clavulanic acid, cefuroxime, and cefixime seemed almost ineffective for UTI infections, regardless of the presence of the *bla*NDM gene. However, meropenem (*p* = 0.000), gentamicin (*p* = 0.000), netilmicin (0.05), and amikacin (*p* = 0.001) appeared to have greater effects over isolates without *bla*NDM (Appendix A).

Among the 23 *blaVIM-positive* isolates, all were resistant to amoxicillin-clavulanic acid and cefuroxime, 96% to cefixime and cefepime, 87% to aztreonam, imipenem and netilmicin, 83% to gentamicin, 65% to amikacin and 39% to meropenem. However, *blaVIM-negative* isolates also showed similar resistance levels to amoxicillin-clavulanic acid, aztreonam, cefuroxime, cefixime, and cefepime, but were more susceptible to imipenem (*p* = 0.17), meropenem (*p* = 0.000), gentamicin (*p* = 0.000), netilmicin (*p* = 0.18) and amikacin (*p* = 0.06). UTI isolates were most susceptible to meropenem (*p* = 0.000) (Appendix A).

Among the three ESBL genes analyzed, *bla*OXA showed significantly higher statistical associations with phenotypic resistance to amoxicillin-clavulanic acid, aztreonam, and cefuroxime sodium (*p* values 0.024, 0.000, and 0.012, respectively, Table 2). However, carbapenem resistance is lower in these strains compared to other CROs. No statistical associations were detected with *bla*TEM and *bla*SHV with phenotypic resistance to the beta-lactam antibiotics tested (Table 2). Phenotypic co-resistance of MBL-carrying isolates was identified. MBL-positive isolates showed 71.2% co-resistance to gentamicin, which was significantly higher compared to that of the MBL-negative isolates (*p* = 0.001). Similarly, phenotypic amikacin co-resistance was observed significantly higher in MBL-positive bacteria (69.5%, *p* = 0.029). Further, the MBL-producing isolate showed 71.2% co-resistance against the monobactam group antibiotic, aztreonam.

The associations of MIC values for meropenem were further analyzed for the two MBL-carrying diarrheal and UTI pathogens. The MIC levels for diarrheal isolates carrying MBL genes (*bla*NDM-1 and *bla*VIM) were almost equal to those for pathogens without MBL genes (Table 3). However, the UTI isolates with the carriage of *bla*NDM-1 and *bla*VIM showed significantly higher meropenem MIC (Table 3).

### 3.6. Phenotype Resistance for Co-Occurance of MBL Genes

Among the 370 bacterial isolates in our study, 60 carried at least one of the two MBL genes (*bla*NDM-1 or *bla*VIM). Most of these isolates carried only one type of gene, although some carried both genes. Co-existing MBL genes were found in seven (13%) bacterial isolates, whereas the other fifty-three (87%) isolates carried only one MBL gene. Only *bla*NDM-1 carrying five isolates showed resistance to all the beta-lactam and aminoglycoside antibiotics tested, while the other 10 showed sensitivity to a wide variety of these antibiotics (Table 4).

MAR index values ranged from 0 (absolute sensitivity) to 1 (absolute resistance), based on the ten antibiotics tested by disk diffusion. Only *bla*VIM-containing isolates exhibited a similar pattern of resistance and sensitivity. MAR index values for *bla*VIM-positive pathogens were 0.2–1.0. In contrast, when *bla*VIM and *bla*NDM-1 co-exist, two isolates appeared sensitive to meropenem and other antibiotics, and the other six isolates were completely resistant to all the antibiotics tested (Table 4). The MAR index value of one isolate was 0.4 and for all the isolates the co-carrying of two MBL genes was very high, 0.9 to 1.0 (Appendix A).

## 4. Discussion

Our study collected two types of samples (diarrheal and UTI). For diarrheal samples, the average pathogen detection rate of bacteria was very high (95%). Different bacteria were isolated from the diarrheal samples, of which Escherichia coli was the predominant bacteria, similar to a study in Nepal [65] and among diarrheal samples from India [66]. For UTI samples, the overall pathogen detection rate was 94.67%. The major etiologic agent found was *Klebsiella* spp., which accounted for up to one-third of the total UTI isolates, with *Escherichia coli* as the second most-prevalent bacteria found. This contrasts with other published studies [67,68], and we will be investigating this further.

Regarding the prevalence of two MBL genes in diarrheal and UTI samples in Bangladesh, the detection rate of *bla*NDM-1 and *bla*VIM genes was almost equal in both samples, at approximately ten percent. This rate is lower, though, than those found in India and Northeast Iran [69,70]. These variations may result from sample size differences, bacteria types, bacteria from different critical anatomical sites, and antibiotic use patterns. Currently, the prevalence of MBL genes is low in Bangladesh compared to other ESBL genes, probably because the use of carbapenem antibiotics is still limited, although growing. However, these genes spread rapidly and pose an appreciable threat to human health in Bangladesh, exacerbated currently by a high consumption of antibiotics from the ‘Watch’ list [10,11]. Overall, numerous issues in managing infections by MBL producers remain unresolved in Bangladesh due to a lack of clinical experience and contradictory epidemiological data. These need to be addressed, going forward, as part of the National Action Plan in Bangladesh to reduce AMR [71].

This study reported the association of MBL genes with phenotypic antibiotic resistance. In the case of diarrheal isolates, almost three quarters of the *bla*NDM-1 positive isolates showed resistance to all the beta-lactam antibiotics tested; however, there was less resistance to aminoglycosides. Having said this, *bla*NDM-1 negative isolates also showed a similar resistance rate to aztreonam, cefuroxime, cefixime, cefepime, gentamicin, and netilmicin. However, in the case of amoxicillin + clavulanic acid, imipenem, meropenem, and amikacin, the resistance rate was much lower in *bla*NDM-1 negative isolates compared to *bla*NDM-1 positive isolates. Encouragingly, *bla*NDM-1 negative isolates were still susceptible to meropenem.

In the case of UTI isolates, all *bla*NDM-1 positives were fully resistant to amoxicillin-clavulanic acid, cefuroxime, cefixime, cefepime, gentamicin, netilmicin, and amikacin and almost entirely resistant to imipenem and meropenem. In addition, the *bla*NDM-1 negative UTI isolates also showed a similar level of resistance to all the beta-lactam antibiotics such as *bla*NDM-1 positives. except meropenem. However, they showed a much lower level of resistance to aminoglycosides. A substantial correlation between MDR and the presence of ESBL genes was also found by Manandhar et al. [72]. A similar association was also seen for the *bla*VIM gene and phenotypic resistance to imipenem, meropenem, and the three aminoglycosides. The resistance rate to meropenem in our study was very low. As a result, appropriate ASPs need to be urgently instigated in ambulatory care in Bangladesh to prevent resistance development.

When the *bla*NDM-1 or *bla*VIM genes were present, they displayed decreased sensitivity to every antibiotic tested. These findings are similar to those of other published studies [44,73]. However, the coexistence of these two genes showed increased resistance to beta-lactam and aminoglycoside antibiotics. Bacteria can accumulate a number of genes, due to the co-resistance phenomena. The coexistence of multiple genes in the same bacteria results in enhanced resistance to various antimicrobials [44,74]. In this study, when both MBL genes coexisted, they were resistant to all the β-lactam and aminoglycosides tested except for meropenem, in only one isolate. According to molecular structural homology, MBL belongs to group B beta-lactamases (Ambler’s classification), which exhibit a wide-spectrum hydrolysis of all beta-lactam antibiotics except aztreonam (monobactam) [75]. Strangely, our study demonstrated aztreonam to be ineffective against over 70% of the isolates carrying MBLs. This situation can be explained by the reported coexistence of ESBL genes and MBL genes within the same isolates. Studies reported that group A ESBL genes such as *bla*TEM, *bla*SHV, and *bla*CTX-M-type can hydrolyze monobactams such as aztreonam and late-generation cephalosporins [76,77]. In a separate analysis, our study showed significant higher resistance of aztreonam in the presence of overall ESBL genes (*p* = 0.043). *bla*KPC, a group A carbapenemase that has not been investigated in this study, may coexist with the MBL genes and impart aztreonam resistance. A statistically significant association of the group D oxacillinase gene (*bla*OXA) was noticed in aztreonam resistance, and substantial coexistence of *bla*OXA with MBL genes was found. Altogether, the coexistence of group A ESBL, group A carbapenemase, and the group D oxacillinase may explain and justify why MBL-positive isolates exhibited co-resistance to aztreonam (monobactam) [75,77]. A very high MAR value in the isolates of coexistence of the MBL genes and other ARGs will require further investigation into the carriage of different integron classes that confer higher AMR acquisition and spreading capacity [78]. Empirical antibiotics are traditionally used to treat uncomplicated UTIs in Bangladesh, which may influence higher resistance via the coexistence of MBL genes. Consequently, again, appropriate educational and ASP activities are urgently needed among healthcare professionals and patients to reduce resistance development to important antibiotics [55,79,80,81].

Overall, the phenotypic antibiotic-resistance patterns of the isolates in our study were seen with seven beta-lactam and three aminoglycoside antibiotics. Bacteria isolated from diarrheal samples showed varying degrees of phenotypic resistance to the various antibiotics studied, with the diarrheal isolates most sensitive to meropenem. Similar types of sensitivity of diarrheal isolates to meropenem were found in a study in India [66]. However, much lower resistance to imipenem and meropenem antibiotics was observed among *E. coli* isolates in a hospital-based study in China [82]. Due to their widespread use in treating UTIs and other illnesses, these antibiotics may have higher levels of resistance [83]. Clinicians have begun to rely more on carbapenems, due to the decreased potency of cephalosporins. From this study, we observed that meropenem was one of the most effective β-lactam antibiotics available, and needs to be preserved, going forward, with suitable ASPs. These isolates also showed good susceptibility to gentamicin and amikacin antibiotics. However, an earlier study in Bangladesh found that 15% of Gram-negative UTI isolates were resistant to imipenem [8], which was five times lower than that seen in our research, illustrating an upward trend in carbapenem resistance in Bangladeshi UTI bacteria, which again needs to be urgently addressed. Other studies have also revealed a rising trend of carbapenem resistance [16]. Imipenem and meropenem are some of the few effective antibiotics for treating various infections. Consequently, the rapid increase in resistance against these antibiotics is a genuine concern, leading to their designation as ‘Watch’ antibiotics; therefore, they should be subject to ASPs to limit inappropriate prescribing and dispensing [3,14,84]. We will continue to monitor the situation in Bangladesh.

We are aware of several limitations of this study. Risk behavior data from diarrheal and UTI patients was not studied. In addition, we used a cross-sectional research approach without follow-up, as we wanted to initially document the extent of resistant pathogens in these common conditions in ambulatory care in Bangladesh to help guide future activities, with ambulatory care accounting for up to 95% of human antibiotic use in LMICs [85]. The sample size was also relatively small, with only two MBL genes, three ESBL genes, and a small number of β-lactam and aminoglycoside antibiotics assessed. Ideally, MIC should be evaluated separately for each AST determination by the disk-diffusion method. Considering our resource limitation, we only evaluated the MIC of meropenem to measure the phenotypic carbapenem resistance. However, internal validity was maintained in this study by replicating independent tests as required. Despite these limitations, we believe this study has elucidated significant diarrheal and UTI pathogens, and their carriage of metallo-beta-lactamase genes, as well as co-resistance and the genotypic–phenotypic relationship with antibiotic resistance, providing future guidance to all key stakeholder groups in Bangladesh.

## 5. Conclusions and Next Steps

In conclusion, our study showed that meropenem is the most efficacious antibiotic for diarrheal and UTI pathogens among patients with these conditions in Bangladesh. The study also reported the emergence of MBL genes in Bangladeshi diarrheal and UTI pathogens. The combination of these genes exerted more resistance than they did separately, which needs to be addressed going forward. When only one MBL gene was present, absolute resistance to the carbapenem group of antibiotics was not conferred; however, in conjunction, they were resistant to all tested antibiotics, even to meropenem, except in one case.

Potential activities to reduce AMR include refining the curricula of all key healthcare professionals, including those managing pharmacies and drug stores in Bangladesh, to make them more aware of the importance of appropriate antibiotic utilization. We have seen in other countries that improved education of pharmacists and drug store personnel, including guidelines, can improve appropriate management, alongside educating prescribers, especially in ambulatory care. Alongside this, patients should be educated to reduce antibiotic requests to treat diarrhea in children and UTIs. The instigation of ASPs can help in this regard, especially following the launch of the AWaRe book giving treatment guidance across a range of infectious diseases seen in ambulatory care coupled with improved surveillance of local resistance patterns. These activities are essential if the Government of Bangladesh is to meet the objectives of the NAP. We will be following up on this in future studies.

## Figures and Tables

**Figure 1 microorganisms-12-01589-f001:**
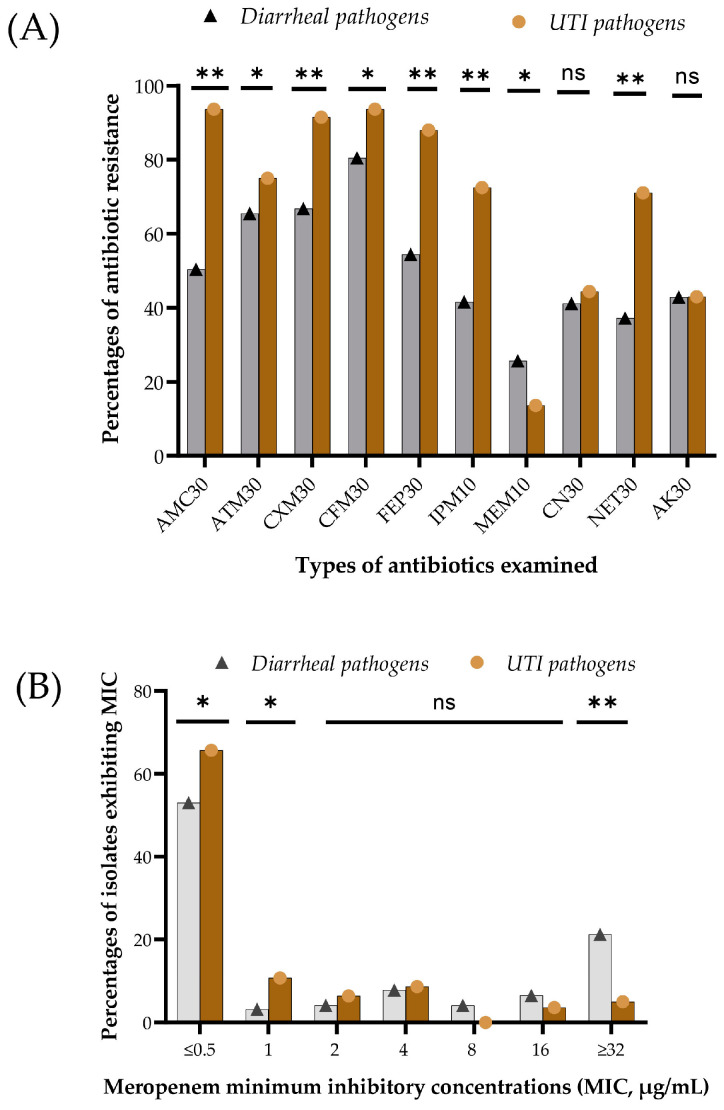
Phenotypic antibiotic resistance in diarrheal and UTI isolates. (**A**) Antibiotic susceptibilities were evaluated largely by the standard disk-diffusion methods for both diarrheal and UTI pathogens against amoxicillin-clavulanic acid (AMC 30 µg), aztreonam (ATM 30 µg), cefuroxime sodium (CXM 30 µg), cefixime (CFM 30 µg), cefepime (FEP 30 µg), imipenem (IMP 10 µg), meropenem (MEM 10 µg), gentamicin (CN 30 µg), netilmicin (NET 30 µg), and amikacin (AK 30 µg). ‘ns’ indicates no statistical differences (*p* > 0.05), ‘*’ means statistical differences were observed (*p* = 0.05 to 0.01) and ‘**’ indicates highly statistical differences were found. (**B**) Differential meropenem minimum inhibitory concentrations (MICs) of diarrheal and UTI isolates are shown. Percentages of bacteria at each MIC level were calculated. The *X*-axis indicates each of the MIC points, and the *Y*-axis represents the percentages of isolates at a particular MIC. *p*-values were calculated from the differences in MIC values between the two groups of clinical pathogens. ‘ns’ shows no statistical differences (*p* > 0.05), ‘*’ means significant statistical differences (*p* = 0.05 to 0.01), and ‘**’ indicates highly significant statistical differences.

**Table 1 microorganisms-12-01589-t001:** Isolation of diarrheal and urinary-tract-infection pathogens by age of study participants.

Age Group (Years)	Diarrheal Pathogens (n = 228)		Urinary-Tract-Infection Pathogens (n = 142)	
	Frequency	Percentage	Frequency	Percentage
1–10	191 ^a^	83.8	5	3.5
11–20	5	2.2	22	15.5
21–30	8	3.5	50 ^b^	35.2
31–40	10	4.4	19 ^b^	13.4
41–50	0	0	27 ^b^	19.0
51–60	5	2.2	8	5.6
61–70	0	0	9	6.3
71–80	9	3.9	2	1.4
Total	228	100	142	100

^a^ One hundred and ninety one (83.9%) of the 228 diarrheal bacteria were isolated from patients aged <10 years. ^b^ Over 60% (86/142) of the UTI isolates were grown from the 21–50-years age range.

**Table 2 microorganisms-12-01589-t002:** Phenotypic–genotypic association of β-lactamase genes and β-lactam antibiotic resistance.

Phenotypic Susceptibility		Presence of β-Lactamase Genes in Diarrheal and UTI Isolates (n = 370)								
		*bla*TEM, No (%)		*p* Value	*bla*OXA, No (%)		*p* Value	*bla*SHV, No (%)		*p* Value
		Positive (n = 165)	Negative (n = 205)		Positive (n = 97)	Negative (n = 273)		Positive (n = 30)	Negative (n = 340)	
*AMC 30*	Sensitive	50 (30.3)	73 (35.6)	0.318	23 (23.7)	100 (36.6)	**0.024**	13 (43.3)	110 (32.4)	0.230
	Resistant	115 (69.7)	132 (64.4)		74 (76.3)	173 (64.4)		17 (56.7)	230 (67.6)	
*ATM 30*	Sensitive	53 (32.1)	63 (30.7)	0.822	16 (16.5)	100 (36.6)	**0.000**	12 (40.0)	104 (30.6)	0.306
	Resistant	112 (67.9)	142 (69.3)		81 (83.5)	173 (64.4)		18 (60.0)	236 (69.4)	
*CXM 30*	Sensitive	38 (23.0)	51 (24.9)	0.715	14 (14.4)	75 (27.5)	**0.012**	7 (23.3)	82 (24.1)	1.00
	Resistant	127 (77.0)	154 (75.1)		83 (85.6)	198 (72.5)		23 (76.7)	258 (75.9)	
*CFM 30*	Sensitive	28 (17.0)	27 (13.2)	0.378	10 (10.3)	45 (16.5)	0.183	9 (30.0)	46 (13.5)	**0.028**
	Resistant	137 (83.0)	178 (86.7)		87 (89.7)	228 (83.5)		21 (70.0)	294 (86.5)	
*FEP 30*	Sensitive	57 (34.5)	65 (31.7)	0.580	23 (23.7)	99 (36.3)	**0.024**	8 (26.7)	114 (33.5)	0.545
	Resistant	108 (65.5)	140 (68.3)		74 (76.3)	174 (63.7)		22 (73.3)	226 (66.5)	
*IMP 10*	Sensitive	83 (50.3)	90 (43.9)	0.249	38 (39.2)	135 (49.5)	0.097	12 (40.0)	161 (47.4)	0.454
	Resistant	82 (49.7)	115 (56.1)		59 (60.9)	138 (50.5)		18 (60.0)	179 (52.6)	
*MEP 10*	Sensitive	130 (79.8)	161 (78.5)	0.798	77 (80.2)	214 (78.7)	0.884	27 (90.0)	264 (78.1)	0.161
	Resistant	33 (20.2)	44 (21.5)		19 (19.8)	58 (21.3)		3 (10.0)	74 (21.9)	

NB: *p* values of statistically significant associations are shown in bold. AMC 30 = amoxycillin-clavulanic acid 30 µg; ATM 30 = aztreonam 30 µg; CXM 30 = cefuroxime sodium 30 µg; CFM 30 = cefixime 30 µg; FEP 30 = cefepime 30 µg; IMP 10 = imipenem 10 µg; MEM 10 = meropenem 10 µg.

**Table 3 microorganisms-12-01589-t003:** Association of the minimum inhibitory concentration of meropenem with the presence of *bla*NDM-1 and *bla*VIM in diarrheal bacteria and uropathogens.

MIC Values (µg/mL)	Diarrheal Pathogens				Urinary-Tract-Infection Pathogen			
	*bla*NDM-1 Carriage		*bla*VIM Carriage		*bla*NDM-1 Carriage		*bla*VIM Carriage	
	Yes (n = 12) ^∆^	No (n = 216)	Yes (n = 12)	No (n = 216)	Yes (n = 13)	No (n = 129)	Yes (n = 23)	No (n = 119)
≤0.5	6 (50)	109 (55)	7 (58)	108 (55)	2 (15)	90 (71)	12 (52)	80 (68)
1.0	1 (8)	7 (3.5)	1 (8)	7 (3.6)	1 (7.7)	14 (11)	1 (4.3)	14 (12)
2.0	0 (0)	9 (4.5)	0 (0)	9 (4.6)	0 (0)	9 (7.1)	0 (0)	9 (7.7)
4.0	0 (0)	17 (8.6)	1 8)	16 (8.1)	6 (46)	6 (4.7)	7 (30)	5 (4.3)
8.0	1 (8)	14 (7.1)	0 (0)	14 (7.1)	1 (7.7)	4 (3.1)	0 (0)	5 (4.3)
16	1 (8)	1 (0.5)	0 (0)	2 (1)	0 (0)	2 (1.6)	0 (0)	2 (1.7)
≥32	3 (25)	41 (21)	3 (25)	41 (21)	3 (23)	2 (1.6)	3 (13)	2 (1.7)

^∆^, number (percentage).

**Table 4 microorganisms-12-01589-t004:** Phenotype resistance associated with *bla*NDM-1, *bla*VIM, and co-occurrence of both MBL genes.

Types of MBL Gene Carriage ^a^	Isolate ID	Phenotypic Susceptibility Assessment ^b^								
		AMC 30	CXM 30	CFM 30	FEP 30	IMP 10	MRP 10	CN 30	NET 30	AK 30
*bla*NDM-1	PBD20	R	R	R	R	R	R	S	S	R
	PBD24	R	S	S	S	S	S	S	S	R
	PBD78	R	S	R	S	S	S	S	S	S
	PBD78C1	R	R	R	R	R	R	S	S	S
	PBD78C2	R	R	R	R	R	R	S	S	S
	PBD86	S	S	S	S	S	S	S	S	S
	PBD86C1	R	R	R	R	R	R	R	R	R
	PBD86C2	R	R	R	R	R	R	R	R	R
	PBD103	S	R	R	S	S	S	R	S	R
	UJ8	R	R	R	R	R	S	R	R	S
	UJ18C2	R	R	R	R	R	R	R	R	R
	UJ29	R	R	R	R	R	R	R	R	R
	UJ49	R	R	R	R	R	R	R	R	R
	UJ56C1	R	R	R	R	S	R	R	R	R
	UJ73	R	R	R	R	R	S	R	R	R
*bla*VIM	PBD5	R	S	R	R	R	R	S	S	S
	PBD8	R	R	R	S	R	S	S	S	S
	PBD40	S	R	R	S	R	S	S	S	S
	PBD53C2	R	R	S	S	R	S	S	S	R
	PBD55	S	S	S	S	S	R	R	S	S
	PBD77	S	S	S	S	S	S	R	R	S
	PBD80C2	S	R	R	R	S	S	R	S	S
	PBD81	R	R	R	S	S	S	S	S	S
	PBD87	R	R	R	R	R	R	R	R	R
	UJ15	R	R	R	R	R	R	R	R	S
	UJ19	R	R	R	R	R	R	R	R	R
	UJ39	R	R	R	R	R	S	R	R	R
	UJ57	R	R	R	R	R	S	R	R	R
	UJ59	R	R	R	R	R	S	R	R	R
	UJ62	R	R	R	R	R	S	R	R	R
	UJ75	R	R	R	R	R	S	R	R	S
	UJ76C1	R	R	R	R	R	S	R	R	R
	UJ77	R	R	R	R	R	S	R	R	S
	UJ78C2	R	R	R	R	R	S	R	R	S
	UJ98	R	R	R	R	S	R	S	S	S
	UJ108C1	S	R	S	S	R	S	S	S	S
	UJ112	R	R	R	R	R	S	R	R	R
	UJ116	R	R	R	R	S	S	R	R	R
	UJ120	R	R	R	R	R	S	R	R	R
	UJ123	R	R	R	R	R	S	S	S	S
*bla*NDM-1 + *bla*VIM	PBD048	S	S	R	S	R	S	R	S	S
	UJ48	R	R	R	R	R	R	R	R	R
	UJ79C1	R	R	R	R	R	R	R	R	R
	UJ79C2	R	R	R	R	R	R	R	R	R
	UJ87C1	R	R	R	R	R	R	R	R	R
	UJ87C2	R	R	R	R	R	S	R	R	R
	UJ88	R	R	R	R	R	R	R	R	R
	UJ90	R	R	R	R	R	R	R	R	R

^a^ Metallo-β-lactamase (MBL) genes; ID = Identity; AMC 30 = amoxicillin-clavulanic acid 30 µg; CXM 30 = cefuroxime 30 µg; CFM 30 = cefixime 30 µg; FEP 30 = cefepime 30 µg; IMP 10 = imipenem 10 µg; MRP 10 = meropenem 30 µg; CN 30 = gentamicin 30 µg; NET 30 = netilmicin 30 µg; AK 30 = amikacin 30 µg; R = resistant; S = sensitive; *bla*NDM-1 = New Delhi metallo-beta lactamase-1; *bla*VIM = verona integron-encoded metallo-beta-lactamase; ^b^ Phenotypic susceptibility assessment to determine the effectiveness of various antibiotics and evaluate how the microorganisms respond against chosen antbiotics.

## Data Availability

Additional data are available from the corresponding author upon reasonable request.

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
