# Peer review of "Resistance and Co-Resistance of Metallo-Beta-Lactamase Genes in Diarrheal and Urinary-Tract Pathogens in Bangladesh"

_microorganisms, 2024, doi:10.3390/microorganisms12081589_

Round 1

Reviewer 1 Report (Previous Reviewer 2)

Comments and Suggestions for Authors

This is an interesting study on the epidemiology of resistance genes in diarrhoeal and urinary tract pathogens collected in a large hospital of the capital of Bangladesh.

The epidemiology reported is vastly different than its neighboring country India, where MBL are the prevalent species. These differences and their underlying causes must be elaborated upon.

It is mentioned that OXA positive strains carry higher b-lactam resistance. This should be rephrased as carbapenem resistance is lower in these strains compared to other CROs.

How is the aztreonam resistance explained in MBL strains?(co-existence of other mechanisms of resistance?)

Co-resistance data in CROs and theory of developing should be analyzed and relevant literature should be added.

Finally, the language of the text needs thorough editing.

Comments on the Quality of English Language

English language needs thorough editing eg

Abstract section:

Line 43 A total of instead of a total 142

Line 42-43 needs rephrasing meaning is not well understood.

Author Response

Please see the file attached.

Reviewer 2 Report (Previous Reviewer 1)

Comments and Suggestions for Authors I think that the manuscript was improved by it needs some other corrections:  
  1. Minor English revision, some mistake throughout the entire manuscript: for example: lane 208 and ESBL genes were obtained previous literature [63,64] ???
  2. The manuscript describes the resistance of isolates from two different anatomical sites, and I think that the best way is to separate these results in two different works as communications, or on the other hand only report the MBL genes characterization without the description of the source of isolates
  3. Table 4 must be improved, very difficult to understand
  4. Figure 2 might be inserted as supplementary material
  5. Lack of the clinical data or previous antibiotic therapy of the subjects, are inpatients or outpatients??or mixed samples?
  I think that major revisions are needed.

Author Response

Please see the file attached.

Round 2

Reviewer 1 Report (Previous Reviewer 2)

Comments and Suggestions for Authors

The manuscript still lacks important information regarding molecular typing of the strains and potential co-existence of mechanisms of resistance. 

Resistance to aztreonam cannot be explained in this way.

Author Response

Reviewer 2 Report (Previous Reviewer 1)

Comments and Suggestions for Authors

Thank you for the revision and improved manuscript

Author Response

Please find the file attached.

This manuscript is a resubmission of an earlier submission. The following is a list of the peer review reports and author responses from that submission.

Round 1

Reviewer 1 Report

Comments and Suggestions for Authors

This work is interesting and it explains and corroborates the severe problem of antimicrobial resistance!!!

But I think that it could be submitted as a communication paper and not as an article, since the data are not enough to describe the health problem of antibiotic resistance and the large spread of resistance genes.

Reduce the number of figures (1 and 2 are not necessary, description in the text) and tables (2 and 3 are not necessary, describe in the text) and describe the most important results in order to submit a communication and not an article.

Reduce figures and tables and describe the most important results

Comments on the Quality of English Language

Minor English revision, some mistakes throughout the manuscript and some content have to be revised

example: lanes 34-36 and 60-61...both difficult to understand

Reviewer 2 Report

Comments and Suggestions for Authors indeed this is an interesting study on a very scientifically important issue. The main question was addressed by the research study, needs however to be further explained. Microbial resistance is a very uptodate important issue and is always relevant. However; the number of the strains analyzed is relatively small to be able to lead to safe and significant conclusions. The authors need to elaborate upon whether other resistance mechanisms besides MBL lactamases were detected in the particular strains which could explain the high resistance to aztreonam for example. Additionally a significant difference in resistance was observed between the imipenem and the meropenem. Were there also IMP enzymes detected?

It is important to mention the exact MICs in the different carbapenems, and correlate these results with the molecular results.

The authors need to explain

a. the significant difference between the MICs in imipenem and the corresponding ones in imipenem and

b. the susceptibility of MBL strains to carbapenems.

The manuscript is not suitable for publication.

Comments on the Quality of English Language

Minor editing eg in Klebsiella pneumonia (the final e has been omitted)